# Oracle Complexity of Single-Loop Switching Subgradient Methods for Non-Smooth Weakly Convex Functional Constrained Optimization

**Yankun Huang**
Department of Business Analytics
University of Iowa
Iowa City, IA 52242
yankun-huang@uiowa.edu

**Qihang Lin**
Department of Business Analytics
University of Iowa
Iowa City, IA 52242
qihang-lin@uiowa.edu

## Abstract

We consider a non-convex constrained optimization problem, where the objective function is weakly convex and the constraint function is either convex or weakly convex. To solve this problem, we consider the classical switching subgradient method, which is an intuitive and easily implementable first-order method whose oracle complexity was only known for convex problems. This paper provides the first analysis on the oracle complexity of the switching subgradient method for finding a nearly stationary point of non-convex problems. Our results are derived separately for convex and weakly convex constraints. Compared to existing approaches, especially the double-loop methods, the switching gradient method can be applied to non-smooth problems and achieves the same complexity using only a single loop, which saves the effort on tuning the number of inner iterations.

## 1 Introduction

Continuous optimization with nonlinear constraints arises from many applications of machine learning and statistics. Examples include Neyman-Pearson classification [62] and learning with fairness constraints [77]. In this paper, we consider the following general nonlinear constrained optimization

$$f^* \equiv \min_{\mathbf{x} \in \mathcal{X}} f(\mathbf{x}) \quad \text{s.t.} \quad g(\mathbf{x}) \leq 0, \tag{1}$$

where $\mathcal{X} \subset \mathbb{R}^d$ is a closed convex set that allows a computationally easy projection operator, $f$ is weakly-convex, $g$ is either convex or weakly convex, and functions $f$ and $g$ are not necessarily smooth. When $g(\mathbf{x}) \equiv \max_{i=1,\dots,m} g_i(\mathbf{x})$, (1) is equivalent to an optimization problem with multiple nonlinear constraints $g_i(\mathbf{x}) \leq 0$ for $i = 1, \dots, m$.

A weakly convex function can be non-convex, so computing an optimal solution of (1) is challenging in general even without constraints. For this reason, theoretical analysis for gradient-based algorithms for non-convex problems mostly focuses on an algorithm's (oracle) complexity for finding an $\epsilon$-stationary solution for (1). When a problem is non-smooth, finding an $\epsilon$-stationary solution is generally difficult even if the problem is convex [47]. Hence, in this paper, we consider finding a nearly $\epsilon$-stationary solution for (1), whose definition will be stated later in Definition 3.2.

In the past decade, there have been many studies on non-convex constrained optimization. However, most of the existing algorithms and their theoretical complexity analysis are developed by assuming $f$ and $g_i$'s are all smooth or can be written as the sum of a smooth and a simple non-smooth functions. A non-exhaustive list of the works under such an assumption includes [34, 80, 81, 55, 44, 45, 43, 52, 63, 53, 50, 12, 22, 9, 21, 40, 51]. Their results cannot be applied to (1) due to non-smoothness in the problem.

37th Conference on Neural Information Processing Systems (NeurIPS 2023).

Non-smoothness is common in optimization in machine learning, e.g., when a non-smooth loss function is applied, but there are much fewer studies on non-smooth non-convex constrained optimization. Under the weak-convexity assumption, an effective approach for solving a non-smooth non-convex problem with theoretical guarantees is the (inexact) proximal point method, where a quadratic proximal term is added to objective and constraint functions to construct a strongly convex constrained subproblem and then a sequence of solutions can be generated by solving this subproblem inexactly and updating the center of the proximal term. Oracle complexity for this method to find a nearly $\epsilon$-stationary has been established by [13, 54, 39] under different constraint qualifications.

The inexact proximal point method is a *double-loop* algorithm where the inner loop is another optimization algorithm for solving the aforementioned strongly convex subproblems. The complexity results in [13, 54, 39] require the inner loop solves each subproblem to a targeted optimality gap. However, the optimality gap is hard to evaluate and thus cannot be used to terminate the inner loop. Although the number of inner iterations needed to achieve the targeted gap can be bounded theoretically, the bound usually involves some constants that are unknown or can only be estimated conservatively. Hence, using the theoretical iteration bound to stop the inner loop usually leads to significantly more inner iterations than needed, making the whole algorithm inefficient. In practices, users often need to tune the number of inner iterations to improve algorithm's efficiency, which is an inconvenience common to most double-loop methods.

On the contrary, a *single-loop* algorithm is usually easier to implement as it does not require tuning the number of inner iterations. Therefore, the **main contribution** of this paper is showing that a single-loop first-order algorithm can find a nearly $\epsilon$-stationary point of (1) with oracle complexity $O(1/\epsilon^4)$, which matches state-of-the-art results obtained only by the double-loop methods [13, 54, 39]. The algorithm we study is the classical *switching subgradient* (SSG) method proposed by Polyak [59] in 1967, which is intuitive and easy to implement but has only been analyzed before in the convex case. We first show that the SSG method has complexity complexity $O(1/\epsilon^4)$ when $g$ is convex. Then, we show that the same complexity result holds when $g$ is weakly convex and appropriate constraint qualification holds. We also present some practical examples where all of our assumptions hold, including a fair classification problem subject to the demographic parity constraint [1]. Our **technical novelty** is inventing a *switching stepsize rule* to accompany the switching subgradient. In particular, we use a fixed stepsize when the solution is updated by the objective's subgradient while use an adaptive Polyak's stepsize [33, 60] when the solution is updated by the constraint's subgradient. This allows us to leverage the local error bound of the constraint function to keep the solution nearly feasible during the algorithm, which prevents the solution from being trapped at an infeasible stationry point. To the best of our knowledge, this paper is the first to establish the complexity of a single-loop first-order method for weakly convex non-smooth nonlinear constrained optimization. In the appendix, we also extend our algorithms and complexity analysis to the stochastic setting.

## 2   Related work

Non-convex constrained optimization has a long history [36, 16, 29, 17, 30, 6, 18] and the interest on this subject has been still growing in the machine learning community because of its new applications such as learning with fairness constraints (see e.g., [77]).

In recent literature, the prevalent classes of algorithms for non-convex constrained optimization include the augmented Lagrangian method (ALM), the penalty method [37, 79, 80, 81, 38, 42, 55, 44, 45, 69, 43, 52, 63, 53, 50, 40, 51], and the sequential quadratic programming method [11, 34, 12, 22, 10, 9, 8, 21]. Besides, an inexact projected gradient method is developed by [15] and a level conditional gradient method is developed by [20]. However, these works all focus on the case where $g$ is smooth and $f$ is either smooth or equals $f_1 + f_2$, where $f_1$ is smooth and non-convex while $f_2$ is non-smooth and convex and has a simple structure that allows a closed-form solution for the proximal mapping $\arg\min_{\mathbf{y}}\{f_2(\mathbf{y}) + \frac{\rho}{2}\|\mathbf{y} - \mathbf{x}\|^2\}$ for any $\mathbf{x}$. There are relatively fewer works on non-convex non-smooth constrained problems. An alternating direction method of multipliers (ADMM) and an ALM are studied by [75] and [78], respectively, for non-convex non-smooth problems with linear constraints while our study considers nonlinear non-smooth constraints. The methods by [20] and [12] can be extended to a structured non-smooth case where $f = f_1 + f_2$ with $f_1$ being smooth non-convex and $f_2 = \max_{\mathbf{y}}\{\mathbf{y}^\top A\mathbf{x} - \phi(\mathbf{y})\}$ with a convex $\phi$, and $g$ has a similar structure. The method by [14] can handle a specific non-smooth non-convex constraint, i.e., $g(\mathbf{x}) = \lambda\|\mathbf{x}\|_1 - h(\mathbf{x})$

where $h$ is a convex and smooth. Compared to these works, our results apply to a more general non-smooth problem without those structure assumptions.

When $f$ and $g$ in (1) are weakly convex and non-smooth, the inexact proximal point method has been studied by [13, 54, 39] under different constraint qualifications and different notions of stationarity. Their complexity analysis utilizes the relationship between the gradient of the Moreau envelope of (1) and the near stationarity of a solution, which is originally used to analyze complexity of subgradient methods for weakly convex non-smooth unconstrained problems [24, 25, 27, 2, 28, 61, 83]. Our analysis utilizes a similar framework. The methods [13, 54, 39] are double-loop while our algorithm only uses a single loop and achieves the same complexity of $O(1/\epsilon^4)$ as them under similar assumptions.

The SSG algorithm is first proposed by Polyak [59]. It has been well-studied for convex problems [58, 7, 49, 67, 72, 73, 65, 68, 66, 3] and quasi-convex problems [67]. This paper provides the first complexity analysis for the SSG method under weak convexity assumption. Non-smooth non-convex optimization has also been studied without weak convexity assumption by [82, 46, 64, 47, 19, 70, 71]. These works analyze the complexity of first-order methods for computing an $(\epsilon, \delta)$-Goldstein approximate stationary point, which is a more general stationarity notation than what we consider here. However, these works only focus on unconstrained problems.

## 3   Algorithm and near stationarity

Let $\| \cdot \|$ be the $\ell_2$-norm. For $h : \mathbb{R}^d \to \mathbb{R} \cup \{+\infty\}$, the subdifferential of $h$ at $\mathbf{x}$ is

$$\partial h(\mathbf{x}) = \left\{ \boldsymbol{\zeta} \in \mathbb{R}^d \mid h(\mathbf{x}') \geq h(\mathbf{x}) + \langle \boldsymbol{\zeta}, \mathbf{x}' - \mathbf{x} \rangle + o(\|\mathbf{x}' - \mathbf{x}\|), \ \mathbf{x}' \to \mathbf{x} \right\},$$

and $\boldsymbol{\zeta} \in \partial h(\mathbf{x})$ is a subgradient of $h$ at $\mathbf{x}$. We say $h$ is $\mu$-*strongly convex* ($\mu \geq 0$) on $\mathcal{X}$ if

$$h(\mathbf{x}) \geq h(\mathbf{x}') + \langle \boldsymbol{\zeta}, \mathbf{x} - \mathbf{x}' \rangle + \frac{\mu}{2} \|\mathbf{x} - \mathbf{x}'\|^2$$

for any $(\mathbf{x}, \mathbf{x}') \in \mathcal{X} \times \mathcal{X}$ and any $\boldsymbol{\zeta} \in \partial h(\mathbf{x}')$. We say $h$ is $\rho$-*weakly convex* ($\rho \geq 0$) on $\mathcal{X}$ if

$$h(\mathbf{x}) \geq h(\mathbf{x}') + \langle \boldsymbol{\zeta}, \mathbf{x} - \mathbf{x}' \rangle - \frac{\rho}{2} \|\mathbf{x} - \mathbf{x}'\|^2$$

for any $(\mathbf{x}, \mathbf{x}') \in \mathcal{X} \times \mathcal{X}$ and any $\boldsymbol{\zeta} \in \partial h(\mathbf{x}')$. We denote the normal cone of $\mathcal{X}$ at $\mathbf{x}$ by $\mathcal{N}_\mathcal{X}(\mathbf{x})$ and the relative interior of $\mathcal{X}$ by $\text{relint}(\mathcal{X})$. We say a point $\mathbf{x}$ is $\epsilon$-feasible if $\mathbf{x} \in \mathcal{X}$ and $g(\mathbf{x}) \leq \epsilon$. Let $\delta_\mathcal{X}(\mathbf{x})$ be the zero-infinity characteristic function of set $\mathcal{X}$, $\text{proj}_\mathcal{X}(\cdot)$ be the projection mapping to $\mathcal{X}$, and $\text{dist}(\mathbf{x}, \mathcal{A}) := \min_{\mathbf{y} \in \mathcal{A}} \|\mathbf{x} - \mathbf{y}\|$ for set $\mathcal{A}$.

We make the following assumptions on (1) throughout the paper.

**Assumption 3.1.** Functions $f$ and $g$ are continuous with $\partial f(\mathbf{x}) \neq \emptyset$ and $\partial g(\mathbf{x}) \neq \emptyset$ on $\mathcal{X}$, and there exists $M$ such that $\|\boldsymbol{\zeta}_f\| \leq M$ and $\|\boldsymbol{\zeta}_g\| \leq M$ for any $\mathbf{x} \in \mathcal{X}$, $\boldsymbol{\zeta}_f \in \partial f(\mathbf{x})$ and $\boldsymbol{\zeta}_g \in \partial g(\mathbf{x})$.

Since (1) is non-convex, finding an $\epsilon$-optimal solution is intractable in general. For a non-convex problem, the target is typically to find a *stationary* point of (1), which is a point $\mathbf{x}^* \in \mathcal{X}$ that satisfies the following Karush-Kuhn-Tucker (KKT) conditions:

$$\boldsymbol{\zeta}_f^* + \lambda^* \boldsymbol{\zeta}_g^* \in -\mathcal{N}_\mathcal{X}(\mathbf{x}^*), \quad \lambda^* g(\mathbf{x}^*) = 0, \quad g(\mathbf{x}^*) \leq 0, \quad \lambda^* \geq 0,$$

where $\lambda^* \in \mathbb{R}$ is a Lagrangian multiplier, $\boldsymbol{\zeta}_f^* \in \partial f(\mathbf{x}^*)$ and $\boldsymbol{\zeta}_g^* \in \partial g(\mathbf{x}^*)$. Typically, an exact stationary point can only be approached by an algorithm at full convergence, which may require infinitely many iterations. Within a finite number of iterations, an algorithm can only generate an $\epsilon$-*stationary* point [13], which is a point $\widehat{\mathbf{x}} \in \mathcal{X}$ satisfying

$$\text{dist}\left( \widehat{\boldsymbol{\zeta}}_f + \widehat{\lambda} \widehat{\boldsymbol{\zeta}}_g, -\mathcal{N}_\mathcal{X}(\widehat{\mathbf{x}}) \right) \leq \epsilon, \quad |\widehat{\lambda} g(\widehat{\mathbf{x}})| \leq \epsilon^2, \quad g(\widehat{\mathbf{x}}) \leq \epsilon^2, \quad \widehat{\lambda} \geq 0, \tag{2}$$

where $\widehat{\lambda} \in \mathbb{R}$ is a Lagrangian multiplier, $\widehat{\boldsymbol{\zeta}}_f \in \partial f(\widehat{\mathbf{x}})$ and $\widehat{\boldsymbol{\zeta}}_g \in \partial g(\widehat{\mathbf{x}})$. However, because $f$ and $g$ are non-smooth, computing an $\epsilon$-stationary point is still challenging even for an unconstrained problem. Nevertheless, under the weak convexity assumption, it is possible to compute a *nearly $\epsilon$-stationary point*, which we will introduce next.

Given $\hat{\rho} \geq 0$, $\tilde{\rho} \geq 0$ and $\mathbf{x} \in \mathcal{X}$, we define a quadratically regularized problem of (1) as

---

**Algorithm 1** Switching Subgradient (SSG) Method by Polyak [59]

---

1: **Input:** $\mathbf{x}^{(0)} \in \mathcal{X}$, total number of iterations $T$, tolerance of infeasibility $\epsilon_t \geq 0$, stepsize $\eta_t$, and a starting index $S$ for generating outputs.
2: **Initialization:** $I = \emptyset$ and $J = \emptyset$.
3: **for** iteration $t = 0, 1, \cdots, T - 1$ **do**
4:     **if** $g(\mathbf{x}^{(t)}) \leq \epsilon_t$ **then**
5:         Set $\mathbf{x}^{(t+1)} = \mathrm{proj}_{\mathcal{X}}(\mathbf{x}^{(t)} - \eta_t \boldsymbol{\zeta}_f^{(t)})$ for any $\boldsymbol{\zeta}_f^{(t)} \in \partial f(\mathbf{x}^{(t)})$ and, if $t \geq S$, $I = I \cup \{t\}$.
6:     **else**
7:         Set $\mathbf{x}^{(t+1)} = \mathrm{proj}_{\mathcal{X}}(\mathbf{x}^{(t)} - \eta_t \boldsymbol{\zeta}_g^{(t)})$ for any $\boldsymbol{\zeta}_g^{(t)} \in \partial g(\mathbf{x}^{(t)})$ and, if $t \geq S$, $J = J \cup \{t\}$.
8:     **end if**
9: **end for**
10: **Output I:** $\mathbf{x}^{(\tau)}$ with $\tau$ randomly sampled from $I$ using $\mathrm{Prob}(\tau = t) = \eta_t / \sum_{s \in I} \eta_s$.
11: **Output II:** $\mathbf{x}^{(\tau)}$ with $\tau$ randomly sampled from $I \cup J$ using $\mathrm{Prob}(\tau = t) = \eta_t / \sum_{s \in I \cup J} \eta_s$.

---

$$\varphi(\mathbf{x}) \equiv \min_{\mathbf{y} \in \mathcal{X}} \left\{ f(\mathbf{y}) + \frac{\hat{\rho}}{2} \|\mathbf{y} - \mathbf{x}\|^2, \ s.t. \ g(\mathbf{y}) + \frac{\tilde{\rho}}{2} \|\mathbf{y} - \mathbf{x}\|^2 \leq 0 \right\}, \tag{3}$$

$$\widehat{\mathbf{x}}(\mathbf{x}) \equiv \arg\min_{\mathbf{y} \in \mathcal{X}} \left\{ f(\mathbf{y}) + \frac{\hat{\rho}}{2} \|\mathbf{y} - \mathbf{x}\|^2, \ s.t. \ g(\mathbf{y}) + \frac{\tilde{\rho}}{2} \|\mathbf{y} - \mathbf{x}\|^2 \leq 0 \right\}. \tag{4}$$

Following the literature on weakly convex optimization [24, 27, 23, 13, 54, 39], we use the value of $\|\widehat{\mathbf{x}}(\mathbf{x}) - \mathbf{x}\|$ as a measure of the quality of a solution $\mathbf{x}$ because it can be interpreted as a stationarity measure. For the purpose of illustration, we assume for now that $\widehat{\mathbf{x}}(\mathbf{x})$ is uniquely defined and there exists a Lagrangian multiplier $\widehat{\lambda} \in \mathbb{R}$ such that the following KKT conditions of (4) holds.

$$\widehat{\boldsymbol{\zeta}}_f + \hat{\rho}(\widehat{\mathbf{x}}(\mathbf{x}) - \mathbf{x}) + \widehat{\lambda}\left(\widehat{\boldsymbol{\zeta}}_g + \tilde{\rho}(\widehat{\mathbf{x}}(\mathbf{x}) - \mathbf{x})\right) \in -\mathcal{N}_{\mathcal{X}}(\widehat{\mathbf{x}}(\mathbf{x})),$$

$$\widehat{\lambda}\left(g(\widehat{\mathbf{x}}(\mathbf{x})) + \frac{\tilde{\rho}}{2} \|\widehat{\mathbf{x}}(\mathbf{x}) - \mathbf{x}\|^2\right) = 0, \quad g(\widehat{\mathbf{x}}(\mathbf{x})) + \frac{\tilde{\rho}}{2} \|\widehat{\mathbf{x}}(\mathbf{x}) - \mathbf{x}\|^2 \leq 0, \quad \widehat{\lambda} \geq 0, \tag{5}$$

where $\widehat{\boldsymbol{\zeta}}_f \in \partial f(\widehat{\mathbf{x}}(\mathbf{x}))$ and $\widehat{\boldsymbol{\zeta}}_g \in \partial g(\widehat{\mathbf{x}}(\mathbf{x}))$. Therefore, as long as $\|\widehat{\mathbf{x}}(\mathbf{x}) - \mathbf{x}\| \leq \epsilon$, we have

$$\mathrm{dist}\left(\widehat{\boldsymbol{\zeta}}_f + \widehat{\lambda}\widehat{\boldsymbol{\zeta}}_g, -\mathcal{N}_{\mathcal{X}}(\widehat{\mathbf{x}}(\mathbf{x}))\right) \leq (\hat{\rho} + \widehat{\lambda}\tilde{\rho})\epsilon, \quad |\widehat{\lambda}g(\widehat{\mathbf{x}}(\mathbf{x}))| = \widehat{\lambda}\tilde{\rho}\epsilon^2/2, \quad g(\widehat{\mathbf{x}}(\mathbf{x})) \leq 0. \tag{6}$$

This means $\widehat{\mathbf{x}}(\mathbf{x})$ is an $O(\epsilon)$-stationary point of the original problem (1) in the sense of (2). Since $\mathbf{x}$ is within an $\epsilon$-distance from $\widehat{\mathbf{x}}(\mathbf{x})$, we call such an $\mathbf{x}$ a *nearly $\epsilon$-stationary point* of (1). We formalize this definition as follows.

**Definition 3.2.** Suppose $\widehat{\mathbf{x}}(\mathbf{x})$ is defined in (4) with $\epsilon \geq 0$. A (stochastic) point $\mathbf{x} \in \mathcal{X}$ is a (stochastic) nearly $\epsilon$-stationary point of (1) if $\mathbb{E}[\|\widehat{\mathbf{x}}(\mathbf{x}) - \mathbf{x}\|] \leq \epsilon$.

Of course, we can claim $\widehat{\mathbf{x}}(\mathbf{x})$ is an $O(\epsilon)$-stationary point of (1) based on (6) only when $\widehat{\lambda}$ in (5) exists and does not go to infinity as $\epsilon$ approaches zero. Fortunately, we can show in Lemmas 4.2 and 5.2 that this is true under some constraint qualifications, which justifies Definition 3.2.

We present the SSG method in Algorithm 1 for finding a nearly $\epsilon$-stationary point of (1). At iteration $t$, we check if the current solution $\mathbf{x}^{(t)}$ is nearly feasible in the sense that $g(\mathbf{x}^{(t)}) \leq \epsilon_t$ for a predetermined tolerance of infeasibility $\epsilon_t$. If yes, the algorithm performs a subgradient step along the subgradient of $f$. Otherwise, the algorithm switches the updating direction to the subgradient of $g$. The algorithm records the iteration indexes of the nearly feasible solutions in set $I$ and other indexes in set $J$. The final output is randomly sampled from the iterates in $I$ or $I \cup J$ with a distribution weighted by the stepsizes $\eta_t$'s. An index $S$ is set so the algorithm only starts to record $I$ and $J$ when $t \geq S$. We study the theoretical *oracle complexity* of Algorithm 1 for finding a nearly $\epsilon$-stationary point, which is defined as the total number of times for which the algorithm queries the subgradient or function value of $f$ or $g$. Our results are presented separately when $g$ is convex and when $g$ is weakly convex.

## 4 Convex constraints

In this section, we first consider a relatively easy case where $f$ is weakly convex but $g$ is convex. In particular, we make the following assumptions in addition to Assumption 3.1 in this section.

**Assumption 4.1.** The following statements hold:

    A. $f(\mathbf{x})$ is $\rho$-weakly convex on $\mathcal{X}$ and $g(\mathbf{x})$ is convex on $\mathcal{X}$.

    B. (Slater's condition) There exists $\mathbf{x}_{\text{feas}} \in \text{relint}(\mathcal{X})$ such that $g(\mathbf{x}_{\text{feas}}) < 0$.

    C. There exists $D \in \mathbb{R}$ such that $\|\mathbf{x} - \mathbf{x}'\| \leq D$ for any $\mathbf{x}$ and $\mathbf{x}'$ in $\mathcal{X}$.

In this section, we choose parameters in (3) such that

$$\hat{\rho} > \rho \text{ and } \tilde{\rho} = 0. \tag{7}$$

Under Assumption 4.1, (7) guarantees that (3) is strictly feasible, its objective function is $(\hat{\rho} - \rho)$-strongly convex and its constraint function is convex, so $\hat{\mathbf{x}}(\mathbf{x})$ in (4) is unique and $\widehat{\lambda}$ in (5) exists. We first present an upper bound of $\widehat{\lambda}$ that is independent of $\mathbf{x}$. The proof is in Section A.1.

**Lemma 4.2.** *Suppose Assumptions 3.1 and 4.1 hold. Given any $\mathbf{x} \in \mathcal{X}$, let $\hat{\mathbf{x}}(\mathbf{x})$ be defined as in (4) with $(\hat{\rho}, \tilde{\rho})$ satisfying (7) and $\widehat{\lambda}$ be the associated Lagrangian multiplier satisfying (5). We have*

$$\widehat{\lambda} \leq \Lambda := (MD + \hat{\rho}D^2)/(-g(\mathbf{x}_{\text{feas}})). \tag{8}$$

For simplicity of notation, we denote $\hat{\mathbf{x}}(\mathbf{x}^{(t)})$ defined in (4) by $\hat{\mathbf{x}}^{(t)}$. Let $\mathbb{E}_\tau[\cdot]$ be the expectation taken only over the random index $\tau$ when the algorithms stop. We present the convergence properties of Algorithm 1 when $\epsilon_t$ and $\eta_t$ are static and diminishing. The proof is provided in Section A.4.

**Theorem 4.3.** *Suppose Assumptions 3.1 and 4.1 hold and $\Lambda$ is as in (8). Let $\hat{\mathbf{x}}(\mathbf{x}^{(t)})$ be defined as in (4) with $(\hat{\rho}, \tilde{\rho})$ satisfying (7) and $\mathbf{x}^{(\tau)}$ is generated by Output I. Algorithm 1 guarantees $\mathbb{E}_\tau[\|\hat{\mathbf{x}}^{(\tau)} - \mathbf{x}^{(\tau)}\|] \leq \epsilon$ and $\mathbb{E}_\tau[g(\mathbf{x}^{(\tau)})] \leq \frac{\epsilon^2(\hat{\rho}-\rho)}{1+\Lambda}$ in either of the following cases.*

*Case I: $S = 0$, $\epsilon_t = \frac{\epsilon^2(\hat{\rho}-\rho)}{1+\Lambda}$, $\eta_t = \frac{2\epsilon^2(\hat{\rho}-\rho)}{5(1+\Lambda)M^2}$ and $T \geq \frac{25M^2D^2(1+\Lambda)^2}{4\epsilon^4(\hat{\rho}-\rho)^2} = O(1/\epsilon^4)$.*

*Case II: $S = T/2$, $\epsilon_t = \frac{5MD}{\sqrt{t+1}}$, $\eta_t = \frac{D}{M\sqrt{t+1}}$ and $T \geq \frac{50M^2D^2(1+\Lambda)^2}{\epsilon^4(\hat{\rho}-\rho)^2} = O(1/\epsilon^4)$.*

Algorithm 1 is single-loop with $O(1)$ oracle complexity per iteration, so its total complexity is just $T = O(1/\epsilon^4)$, which matches the start-of-the-art complexity by [13, 54, 39]. This result can be generalized to the case where there exist additional linear equality constraints $\mathbf{Ax} = \mathbf{b}$. See the extension of Lemma 4.2 in Section A.1 and Remark A.2.

Assumption 4.1C can be relaxed to only require that the feasible set, i.e., $\mathcal{S} = \{\mathbf{x} \in \mathcal{X} \mid g(\mathbf{x}) \leq 0\}$, is bounded instead of $\mathcal{X}$. The same complexity can be achieved by Algorithm 1 by using a switching step size rule similar to $\eta_t$ in Proposition 5.5. This result is provided in Section A.8 but we recommend interested readers to read Section 5 first to get introduced to this special step size.

*Remark* 4.4. Property $\mathbb{E}_\tau[g(\mathbf{x}^{(\tau)})] \leq \frac{\epsilon^2(\hat{\rho}-\rho)}{1+\Lambda}$ in the theorems above is not required by Definition 3.2. By Assumption 3.1A, $\mathbb{E}_\tau[g(\mathbf{x}^{(\tau)})] \leq \mathbb{E}_\tau[g(\hat{\mathbf{x}}^{(\tau)})] + M\mathbb{E}_\tau[\|\hat{\mathbf{x}}^{(\tau)} - \mathbf{x}^{(\tau)}\|] \leq M\epsilon$, which means a nearly $\epsilon$-stationary point must be $O(\epsilon)$-feasible by definition. Property $\mathbb{E}_\tau[g(\mathbf{x}^{(\tau)})] \leq \frac{\epsilon^2(\hat{\rho}-\rho)}{1+\Lambda}$ implies $O(\epsilon^2)$-feasibility for the output, which is even better.

When $g$ is $\mu$-strongly convex with $\mu > 0$, we can show that the complexity of Algorithm 1 is still $O(1/\epsilon^4)$ but one can simply set $\epsilon_t = 0$, which makes $\eta_t$ the only tuning parameter. This makes this single-loop method even more attractive. Due to space limit, we include this result in Section A.5. We also extend our result to the stochastic case in Section A.6 and A.7.

## 5  Weakly convex constraints

Next we consider the case where both $f$ and $g$ are weakly convex but not necessarily convex. Let

$$g_+(\mathbf{x}) = \max\{g(\mathbf{x}), 0\}, \quad \mathcal{L} = \{\mathbf{x} \in \mathcal{X} \mid g(\mathbf{x}) = 0\} \text{ and } \mathcal{S} = \{\mathbf{x} \in \mathcal{X} \mid g(\mathbf{x}) \leq 0\}.$$

We make the following assumptions in addition to Assumption 3.1 in this section.

**Assumption 5.1.** The following statements hold:

A. $f(\mathbf{x})$ and $g(\mathbf{x})$ are $\rho$-weakly convex on $\mathcal{X}$.

B. There exist $\bar{\epsilon} > 0$, $\theta > 0$ and $\bar{\rho} > \rho$ such that, for any $\bar{\epsilon}^2$-feasible solution $\mathbf{x}$, there exists $\mathbf{y} \in \mathrm{relint}(\mathcal{X})$ such that $g(\mathbf{y}) + \frac{\bar{\rho}}{2}\|\mathbf{y} - \mathbf{x}\|^2 \leq -\theta$.

C. $\underline{f} := \inf_{\mathbf{x} \in \mathcal{X}} f(\mathbf{x}) > -\infty$.

Assumption 5.1B is called the uniform Slater's condition by [54]. We will present two real-world examples in Section B.1 that satisfy this assumption, including a fair classification problem under demographic parity constraint, which is one of the applications in our numerical experiments in Section 6.2. In this section, we choose parameters in (3) such that

$$\bar{\rho} \geq \hat{\rho} = \tilde{\rho} > \rho. \tag{9}$$

Under Assumption 5.1, (9) guarantees that (3) is uniformly strictly feasible for any $\epsilon^2$-feasible $\mathbf{x}$, and the objective and constraint functions of (3) are both $(\hat{\rho} - \rho)$-strongly convex, so $\hat{\mathbf{x}}(\mathbf{x})$ is uniquely defined and $\hat{\lambda}$ in (5) exists. In addition, Assumption 5.1 has the following three implications that play important roles in our complexity analysis.

First, $\hat{\lambda}$ in (5) can be bounded by a constant independent of $\mathbf{x}$ and $\epsilon$ as long as $\mathbf{x}$ is $\epsilon^2$-feasible with $\epsilon \leq \bar{\epsilon}$. This result is similar to Lemma 1 by Ma et al. [54] except that they require $\mathcal{X}$ to be bounded but we do not. The proof is given in Section B.2.

**Lemma 5.2.** *Suppose Assumptions 3.1 and 5.1 hold. Given any $\epsilon^2$-feasible $\mathbf{x}$ with any $\epsilon \leq \bar{\epsilon}$, let $\hat{\mathbf{x}}(\mathbf{x})$ defined as in (4) satisfying (9) and $\hat{\lambda}$ is the associated Lagrangian multiplier satisfying (5). We have*

$$\|\hat{\mathbf{x}}(\mathbf{x}) - \mathbf{x}\| \leq M/\hat{\rho} \quad and \quad \hat{\lambda} \leq \Lambda' := 2M/\sqrt{2\theta(\hat{\rho} - \rho)}. \tag{10}$$

Second, the subgradient of the constraint function $g(\mathbf{x}) + \delta_{\mathcal{X}}(\mathbf{x})$ on $\mathcal{L}$ is uniformly away from the origin. The proof is provided in Section B.2.

**Lemma 5.3.** *Suppose Assumptions 3.1 and 5.1 hold. It holds for any $\mathbf{x} \in \mathcal{L}$ that*

$$\min_{\boldsymbol{\zeta}_g \in \partial g(\mathbf{x}), \mathbf{u} \in \mathcal{N}_{\mathcal{X}}(\mathbf{x})} \|\boldsymbol{\zeta}_g + \mathbf{u}\| \geq \nu := \sqrt{2\theta(\hat{\rho} - \rho)}. \tag{11}$$

Lastly, note that $\mathcal{S}$ is the optimal set of $\min_{\mathbf{x} \in \mathcal{X}} g_+(\mathbf{x})$, which is a $\rho$-weakly convex non-smooth optimization problem with an optimal value of zero. Lemma 5.3 implies that $g_+(\mathbf{x})$ is sharp near the boundary of $\mathcal{S}$, meaning that $g_+(\mathbf{x})$ satisfies a linear error bound in an $O(1)$-neighborhood of $\mathcal{S}$. A similar result for a convex $g_+$ is given in Lemma 1 in [76]. In the lemma below, we extend their result for a weakly convex $g_+$. The proof is in Section B.2 and the second conclusion is directly from [26].

**Lemma 5.4.** *Suppose Assumptions 3.1 and 5.1 hold. It holds for any $\mathbf{x}$ satisfying $\mathrm{dist}(\mathbf{x}, \mathcal{S}) \leq \frac{\nu}{\rho}$ that*

$$(\nu/2)\mathrm{dist}(\mathbf{x}, \mathcal{S}) \leq g_+(\mathbf{x}). \tag{12}$$

*Moreover, $\nu \leq 2M$ and $\min_{\mathbf{x} \in \mathcal{X}} g_+(\mathbf{x})$ has no stationary point satisfying $0 < \mathrm{dist}(\mathbf{x}, \mathcal{S}) < \frac{\nu}{\rho}$.*

Since $g$ is non-convex, Algorithm 1 may not even find a nearly feasible solution if $\mathbf{x}^{(t)}$ is trapped at a stationary point of $g$ with $g(\mathbf{x}) > 0$, that is, a sub-optimal stationary point of $\min_{\mathbf{x} \in \mathcal{X}} g_+(\mathbf{x})$. Fortunately, the second conclusion of Lemma 5.4 indicates that this situation can be avoided by keeping $\mathrm{dist}(\mathbf{x}^{(t)}, \mathcal{S}) = O(\epsilon^2)$ during the algorithm. To do so, we start with $\mathbf{x}^{(0)} \in \mathcal{S}$ and use $\epsilon_t = O(\epsilon^2)$ in Algorithm 1. Moreover, we apply a switching stepsize rule that sets $\eta_t = O(\epsilon^2)$ when $t \in I$ and $\eta_t = g(\mathbf{x}^{(t)})/\|\boldsymbol{\zeta}_g^{(t)}\|^2$ when $t \in J$, the latter of which is known by the Polyak's stepsize [33, 60]. This way, when $g(\mathbf{x}^{(t)}) \leq \epsilon_t$, (12) ensures $\mathrm{dist}(\mathbf{x}^{(t)}, \mathcal{S}) = O(\epsilon^2)$. When $g(\mathbf{x}^{(t)}) > \epsilon_t$, (12) ensures $\mathrm{dist}(\mathbf{x}^{(t)}, \mathcal{S})$ Q-linearly converges to zero [26], which also guarantees $\mathrm{dist}(\mathbf{x}^{(t)}, \mathcal{S}) = O(\epsilon^2)$. As a result, we have $g(\mathbf{x}^{(t)}) \leq \epsilon^2$ for any $t$, so problem (3) with $\mathbf{x} = \mathbf{x}^{(t)}$ and $(\hat{\rho}, \tilde{\rho})$ satisfying (9) will be strictly feasible according to Assumption 5.1B. This finding is given in the proposition below with its proof in Section B.3.

**Proposition 5.5.** *Suppose Assumptions 3.1 and 5.1 hold and $\epsilon \leq \bar{\epsilon}$. Also, suppose $\mathbf{x}^{(t)}$ is generated by Algorithm 1 using $\mathbf{x}^{(0)} \in \mathcal{S}$, $\epsilon_t = \frac{\nu}{4} \min\{\epsilon^2/M, \nu/(4\rho)\}$ and*

$$\eta_t = \begin{cases} \frac{\nu}{4M^2} \min\{\epsilon^2/M, \nu/(4\rho)\} & \text{if } t \in I \\ g(\mathbf{x}^{(t)})/\|\boldsymbol{\zeta}_g^{(t)}\|^2 & \text{if } t \in J. \end{cases}$$

*Then* $\text{dist}(\mathbf{x}^{(t)}, \mathcal{S}) \leq \min\{\epsilon^2/M, \nu/(4\rho)\}$ *and* $g(\mathbf{x}^{(t)}) \leq \epsilon^2$ *for any* $t \geq 0$. *As a consequence,* $\mathbf{x}^{(t)}$ *is* $\epsilon^2$-*feasible to* (3) *where* $\mathbf{x} = \mathbf{x}^{(t)}$ *and* $(\hat{\rho}, \tilde{\rho})$ *satisfies* (9).

Let $\widehat{\mathbf{x}}^{(t)}$ be $\widehat{\mathbf{x}}(\mathbf{x}^{(t)})$ defined in (4) with $(\hat{\rho}, \tilde{\rho})$ satisfying (9). The complexity result for the case of weakly convex constraints is as follows. The proof can be found in Section B.3.

**Theorem 5.6.** *Under the same assumptions as Proposition 5.5, Algorithm 1 guarantees* $\mathbb{E}_\tau[\|\widehat{\mathbf{x}}^{(\tau)} - \mathbf{x}^{(\tau)}\|] \leq C\epsilon$ *and* $\mathbb{E}_\tau[g(\mathbf{x}^{(\tau)})] \leq \epsilon^2$, *where* $C := 2\sqrt{\frac{1+\Lambda'}{\hat{\rho}-\rho}}$, *if we use Output II and set*

$$S = 0 \text{ and } T \geq \frac{8M^2\left(f(\mathbf{x}^{(0)}) - \underline{f} + 3M^2/(2\hat{\rho})\right)}{\hat{\rho}(1 + \Lambda')\nu\epsilon^2 \min\{\epsilon^2/M, \nu/(4\rho)\}} = O(1/\epsilon^4).$$

This theorem indicates that Algorithm 1 finds a nearly $(C\epsilon)$-stationary point with complexity $O(1/\epsilon^4)$. To obtain a nearly $\epsilon$-stationary point with $\epsilon \leq \bar{\epsilon}$, one only needs to replace $\epsilon$ in $\eta_t$, $\epsilon_t$ and $T$ in this theorem above by $\epsilon/\max\{C, 1\}$. This will only change the constant factor in the $O(1/\epsilon^4)$ complexity. This complexity matches the start-of-the-art complexity by [13, 54, 39].

# 6 Numerical experiments

We demonstrate the performance of the SSG method on two fairness-aware classification problems, which are instances of (1) with convex and weakly convex $g$'s, respectively. We compare with two different double-loop inexact proximal point (IPP) methods [13, 54, 39]. The IPP method approximately solves a strongly convex constrained subproblem in each outer iteration, and we use the SSG method in this paper and the ConEx method in [13] as the solvers (inner loop) because they both have the best theoretical complexity for that subproblem. We use IPP-SSG and IPP-ConEx to denote these two implementations of the IPP method. All experiments are conducted using MATLAB 2022b on a computer with the CPU 3.20GHz x Intel Core i7-8700 and 16GB memory.

## 6.1 Classification problem with ROC-based fairness

Given a feature vector $\mathbf{a} \in \mathbb{R}^d$ and a class label $b \in \{1, -1\}$, the goal in a binary linear classification problem is to learn a model $\mathbf{x} \in \mathbb{R}^d$ to predict $b$ based on the score $\mathbf{x}^\top \mathbf{a}$. Let $\mathcal{D} = \{(\mathbf{a}_i, b_i)\}_{i=1}^n$ be a training set and $\ell(\cdot)$ be a convex non-increasing loss function. Model $\mathbf{x}$ can be learned by solving

$$L^* = \min_{\mathbf{x} \in \mathcal{W}} \left\{ L(\mathbf{x}) := \frac{1}{n} \sum_{i=1}^n \ell(b_i \mathbf{x}^\top \mathbf{a}_i) \right\}, \tag{13}$$

where $\mathcal{W} = \{\mathbf{x} \in \mathbb{R}^d \mid \|\mathbf{x}\| \leq r\}$. Solving (13) may ensure good classification performance of $\mathbf{x}$ but not its fairness. Suppose there exist two additional datasets. One contains the feature vectors of a protected group, denoted by $\mathcal{D}_p = \{\mathbf{a}_i^p\}_{i=1}^{n_p}$, and the other one contains the feature vectors of an unprotected group, denoted by $\mathcal{D}_u = \{\mathbf{a}_i^u\}_{i=1}^{n_u}$. We want to enhance the fairness of $\mathbf{x}$ between these two groups using the ROC-based fairness metric proposed by [74]. Suppose we set a threshold $\theta$ and classify data $\mathbf{a}$ as positive if $\mathbf{x}^\top \mathbf{a} \geq \theta$ and as negative otherwise. The ROC-based fairness measure and its continuous approximation are defined as

$$\max_{\theta \in \Theta} \left| \frac{1}{n_p} \sum_{i=1}^{n_p} \mathbb{I}(\mathbf{x}^\top \mathbf{a}_i^p \geq \theta) - \frac{1}{n_u} \sum_{i=1}^{n_u} \mathbb{I}(\mathbf{x}^\top \mathbf{a}_i^u \geq \theta) \right|$$

$$\approx R(\mathbf{x}) := \max_{\theta \in \Theta} \left| \frac{1}{n_p} \sum_{i=1}^{n_p} \sigma(\mathbf{x}^\top \mathbf{a}_i^p - \theta) - \frac{1}{n_u} \sum_{i=1}^{n_u} \sigma(\mathbf{x}^\top \mathbf{a}_i^u - \theta) \right|, \tag{14}$$

where $\sigma(z) = \exp(z)/(1 + \exp(z))$ is the sigmoid function and $\Theta$ is a finite set of thresholds. If the value of this measure is small, model $\mathbf{x}$ produces similar predicted positive rates for the protected and unprotected groups on various $\theta$'s, indicating the fairness of the model. To obtain a fair $\mathbf{x}$, we balance (13) and (14) by solving

$$\min_{\mathbf{x} \in \mathcal{W}} R(\mathbf{x}) \text{ s.t. } L(\mathbf{x}) \leq L^* + \kappa, \tag{15}$$

Table 1: Information of the datasets. Groups are males VS females in a9a, users with age within $[25, 60]$ VS outside $[25, 60]$ in bank, and caucasian VS non-caucasian in COMPAS.

| DATASETS | $n$ | $d$ | LABEL | GROUPS |
|---|---|---|---|---|
| A9A | 48,842 | 123 | INCOME | GENDER |
| BANK | 41,188 | 54 | SUBSCRIPTION | AGE |
| COMPAS | 6,172 | 16 | RECIDIVISM | RACE |

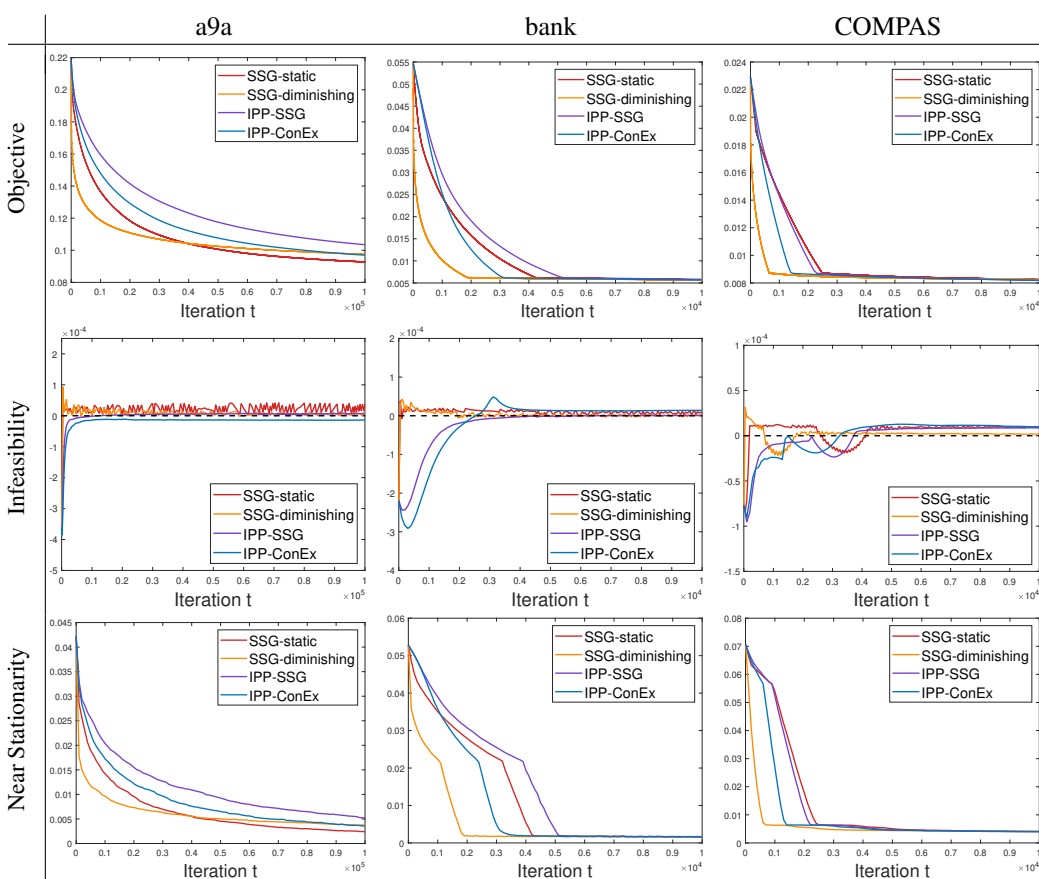

Figure 1: Performances vs number of iterations on classification problems with ROC-based fairness.

where $\kappa$ is a slackness parameter indicating how much we are willing to increase the classification loss in order to reduce $R(\mathbf{x})$ to obtain a more fair model. Problem (15) is an instance of (1) satisfying Assumptions 3.1 and 4.1 with $\rho = \beta$ where $\beta$ is defined in (77) in Section B.1.

We solve (15) on three datasets: *a9a* [41], *bank* [56] and *COMPAS* [4]. The information of these datasets is given in Table 1. We split each dataset into two subsets with a ratio of $2 : 1$. The larger set is used as $\mathcal{D}$ in the constraint and the smaller set is further split into $\mathcal{D}_p$ and $\mathcal{D}_u$ based on a binary group variable listed in Table 1.

In our experiments, we choose $\ell(z) = (1 - z)_+$ and first solve (13) using the subgradient method with a large enough number of iterations to obtain $L^*$ and a solution $\mathbf{x}_{\text{ERM}}$. Then we set $\kappa = 0.001L^*$ and $r = 5\|\mathbf{x}_{\text{ERM}}\|$, and let $\Theta$ consist of 400 points equally spaced between $\min_i \mathbf{x}_{\text{ERM}}^\top \mathbf{a}_i - 0.5(\max_i \mathbf{x}_{\text{ERM}}^\top \mathbf{a}_i - \min_i \mathbf{x}_{\text{ERM}}^\top \mathbf{a}_i)$ and $\max_i \mathbf{x}_{\text{ERM}}^\top \mathbf{a}_i + 0.5(\max_i \mathbf{x}_{\text{ERM}}^\top \mathbf{a}_i - \min_i \mathbf{x}_{\text{ERM}}^\top \mathbf{a}_i)$.

All methods are initialized at $\mathbf{x}_{\text{ERM}}$. We implemented the SSG method with both static and diminishing stepsizes. For the static stepsize, we select $\epsilon_t$ from $\{10^{-6}, 2 \times 10^{-6}, 5 \times 10^{-6}, 10^{-5}\}$ and $\eta_t$ from $\{2 \times 10^{-4}, 5 \times 10^{-4}, 10^{-3}, 2 \times 10^{-3}\}$. For the diminishing stepsize, we set $\epsilon_t = \frac{E_1}{\sqrt{t+1}}$ and $\eta_t = \frac{E_2}{\sqrt{t+1}}$ and select $E_1$ from $\{5 \times 10^{-5}, 10^{-4}, 2 \times 10^{-4}, 5 \times 10^{-4}\}$ and $E_2$ from $\{0.02, 0.05, 0.1, 0.2\}$. We

select the best set of parameters that produces the smallest objective value after 5000 iterations. For IPP, we select $\hat{\rho}$ from $\max\{\rho, 1\} \times \{1, 1.5, 2\}$ for all three datasets, and the proximal point subproblem is approximately solved by SSG and ConEx both with 100 iterations. For IPP-SSG, we apply a static stepsize with $\epsilon_t$ and $\eta_t$ tuned in the same way as SSG. For IPP-ConEx, following the notation in [13], we set $\theta_t = \frac{t}{t+1}$, $\eta_t = c_1(t+1)$ and $\tau_t = \frac{c_2}{t+1}$, and select $c_1$ from $\{20, 50, 100, 200\}$ and $c_2$ from $\{0.002, 0.005, 0.01, 0.02\}$ by the same procedure adopted by SSG.

We report the performances versus number of iterations of all methods on each dataset in Figure 1. The performances versus the used CPU time will be given in Section C in the Appendix. For SSG, the $x$-axis represents the total number of iterations while, for IPP, it represents the total number of inner iterations across all outer iterations. The $y$-axis in each row represents the objective value, infeasibility and near stationarity achieved at each iteration, respectively. To measure near stationarity, we solve (3) with $\mathbf{x} = \mathbf{x}^{(t)}$ and parameters (7) using the SSG method with 2500 iterations and use the last iterate as an approximation of $\hat{\mathbf{x}}(\mathbf{x}^{(t)})$. We make sure that the change of $\|\hat{\mathbf{x}}(\mathbf{x}^{(t)}) - \mathbf{x}^{(t)}\|$ is less than 1% if the number of iterations is increased to 5000. Then we plot $\|\hat{\mathbf{x}}(\mathbf{x}^{(t)}) - \mathbf{x}^{(t)}\|$ as near stationarity in Figure 1. Since computing $\hat{\mathbf{x}}(\mathbf{x}^{(t)})$ with a high precision for each $t$ is time-consuming, we only report near stationarity at 100 equally spaced iterations.

According to Figure 1, the SSG method with a diminishing stepsize has the best performance on all three datasets in the sense that it reduces the objective value and the (approximate) near stationarity measure faster than others while keeping the solutions nearly feasible. However, the SSG method with a static stepsize is not always better than the IPP methods. This is consistent with our theoretical finding that the SSG and IPP methods have the same oracle complexity.

## 6.2 Classification problem with demographic parity

Following the notation in the previous subsection, we consider a binary classification problem with a constraint enforcing demographic parity [1]. The measure of demographic parity and its continuous approximation are

$$\left| \frac{1}{n_p} \sum_{i=1}^{n_p} \mathbb{I}(\mathbf{x}^\top \mathbf{a}_i^p \geq 0) - \frac{1}{n_u} \sum_{i=1}^{n_u} \mathbb{I}(\mathbf{x}^\top \mathbf{a}_i^u \geq 0) \right| \approx R_0(\mathbf{x}) := \left| \frac{1}{n_p} \sum_{i=1}^{n_p} \sigma(\mathbf{x}^\top \mathbf{a}_i^p) - \frac{1}{n_u} \sum_{i=1}^{n_u} \sigma(\mathbf{x}^\top \mathbf{a}_i^u) \right|. \quad (16)$$

Fairness measure $R_0(\mathbf{x})$ is a special case of $R(\mathbf{x})$ with $\Theta = \{0\}$. If $R_0(\mathbf{x})$ is small, model $\mathbf{x}$ produces similar predicted positive rates for the protected and unprotected groups. To obtain a fair $\mathbf{x}$, we balance (13) and (16) by solving

$$\min L(\mathbf{x}) + \lambda \text{SCAD}(\mathbf{x}) \text{ s.t. } R_0(\mathbf{x}) \leq \kappa. \quad (17)$$

Different from (15), the fairness measure is used as the constraint in (17) while the objective function becomes the empirical hinge loss plus the smoothly clipped absolute deviation (SCAD) regularizer [35] for promoting a sparse solution. Here, $\lambda$ is a regularization parameter and

$$\text{SCAD}(\mathbf{x}) := \sum_{i=1}^{d} s(x_i), \quad s(x_i) = \begin{cases} 2|x_i| & 0 \leq |x_i| \leq 1 \\ -x_i^2 + 4|x_i| + 1 & 1 < |x_i| \leq 2 \\ 3 & 2 < |x_i|. \end{cases} \quad (18)$$

We prove in Section B.1 that (17) is an instance of (1) satisfying Assumptions 3.1 and 5.1 with $\rho = \max\{2\lambda, \beta\}$ where $\beta$ is defined in (77) in Section B.1.

We set $\lambda = 0.2$ for all datasets and set $\kappa = 0.005$, $0.02$ and $0.02$ for *a9a*, *bank* and *COMPAS*, respectively. For SSG, we select $\epsilon_t$ from $\{10^{-6}, 2 \times 10^{-6}, 5 \times 10^{-6}, 10^{-5}\}$ and select $\eta_t$ from $\{10^{-4}, 2 \times 10^{-4}, 5 \times 10^{-4}, 7.5 \times 10^{-4}\}$ for $t \in I$ while set $\eta_t = g(\mathbf{x}^{(t)})/\|\boldsymbol{\zeta}_g^{(t)}\|^2$ for $t \in J$. We select the best set of parameters that produces the smallest objective value after 50000 iterations. For IPP, we select $\hat{\rho}$ from $\max\{\rho, 1\} \times \{1, 1.5, 2\}$ and the proximal point subproblem is approximately solved by SSG and ConEx both with 600 iterations. For IPP-SSG, we apply a static stepsize with $\epsilon_t$ and $\eta_t$ tuned in the same way as SSG. For IPP-ConEx, following the notation in [13], we set $\theta_t = \frac{t}{t+1}$, $\eta_t = c_1(t+1)$ and $\tau_t = \frac{c_2}{t+1}$ and select $c_1$ from $\{20, 50, 100, 200\}$ and $c_2$ from $\{0.002, 0.005, 0.01, 0.02\}$ by the same procedure adopted by SSG. Due to the limit of space, we present the performances vs number of iterations and CPU runtime of all methods in Figure 3 and Figure 4 respectively in Section C in the Appendix.

# 7 Conclusion

We study the oracle complexity of the switching subgradient (SSG) method for finding a nearly $\epsilon$-stationary point of a non-smooth weakly convex constrained optimization problem. We show that the complexity of the SSG method matches the best result in literature that is achieved only by double-loop methods. On the contrary, the SSG method is single-loop and easier to implement with reduced tuning effort. This is the first complexity result for a single-loop first-order method for a weakly-convex non-smooth constrained problem.

## Acknowledgements

This work was supported by NSF award 2147253.

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
