# OpenReview forum: "Oracle Complexity of Single-Loop Switching Subgradient Methods for Non-Smooth Weakly Convex Functional Constrained Optimization"
_NeurIPS.cc/2023/Conference — NeurIPS 2023 poster_

### Official Review · Reviewer_DHUQ · 2023-07-05

**Soundness:** 3 good
**Presentation:** 3 good
**Contribution:** 3 good
**Rating:** 6
**Confidence:** 4

**Summary:**

This paper studies switching subgradient method for solving weakly-convex objective (weakly-)convex constrained problems. The oracle complexity for finding a nearly stationary point is derived for both convex constraint case and weakly-convex constrained (with good initialization). Numerical experiments validate the efficiency of the proposed algorithms on classical machine learing tasks.

**Strengths:**

The paper is overall well written. It is the first single-loop first-order method handling a weakly-convex objective (weakly-)convex constrained problems, which is much convenient to implement compared to double-loop algorithms. The theoretical complexity result is clearly stated and competitive with sound proofs.

**Weaknesses:**

No obvious weaknesses and please see the questions part for some detailed comments.

**Questions:**

1. For the weakly-convex constrained case, if we cannot have feasible initial point, does the algorithm SSG converge? If so, what kind of point will it arrive? As we know, it is usually a NP-hard problem to obtain a feasible point for nonconvex constraint. In this case, does the SSG converge to the stationary point satisfying $0\in \partial (f+1_{\mathcal{X}\cap [g\leq0]})(x)$?
2. In Assumption 4.1, it is assumed that the Slater's condition holds. Does it mean that we cannot include the equality constraint $g(x)=0$ for the following analysis?

---

> ### Author Rebuttal · Authors · 2023-08-09
>
> Thank you so much for reviewing our paper and providing valuable feedback that helps us further improve our work!
>
> **Question 1: For the weakly-convex constrained case, if we cannot have feasible initial point, does the algorithm SSG converge? If so, what kind of point will it arrive? As we know, it is usually a NP-hard problem to obtain a feasible point for nonconvex constraint. In this case, does the SSG converge to the stationary point satisfying $0\\in\\partial(f+1_{\mathcal{X}\cap[g\leq0]})(\mathbf{x})$?**
>
> If we don't have a feasible initial point, the SSG method will still converge but it will not necessarily converge to an nearly $\\epsilon$-stationary point under our definition. Instead, it may converge to an infeasible and nearly stationary point of $g$, namely, a point $\\mathbf{x}\in\mathcal{X}$ that satisfies $0\\in\\partial(1_{\mathcal{X}}+g)(\mathbf{x})$ but has $g(\mathbf{x})$ significantly larger than $0$. At this point, the subgradient $\partial 1_{\mathcal{X}\cap[g\leq0]}(\mathbf{x})$ the reviewer conjectured is not defined.
>
> This phenomenon can explained by the design of the SSG algorithm. In fact, if $\mathbf{x}^{(t)}$ is very infeasible so $g(\mathbf{x}^{(t)})>\epsilon_t$, the SSG method will keep using the subgradient of $g$ to update $\mathbf{x}^{(t)}$. Until $g(\mathbf{x}^{(t)})\leq\epsilon_t$, the SSG method is equivalent to the subgradient descend method applied to $\min_{\mathbf{x}\in\mathcal{X}} g(\mathbf{x})$. Since $g$ is non-convex, the subgradient descend method may fail to reduce $g$ to zero so $g(\mathbf{x}^{(t)})\leq\epsilon_t$ may never happen, but the subgradient descend method can at least ensure  $\mathbf{x}^{(t)}$ converge to a nearly stationary point of $g$. Please note that, in this scenario, the subgradient of $f$ is never used so $f$ has zero influence on where $\mathbf{x}^{(t)}$ converges to.
>
> From the explanation above, we also know that non-convex constrained optimization is intractable if the initial solution can be anywhere. Please consider problem $\min_{x} -(x-1)^3 \text{s.t.} (x-1)^3+1\leq 0 $. Here, $x=1$ is an infeasible point but the gradients of the objective and the constraint functions are both zero at this point. In other words, if $x=1$ is the initial solution, all gradient-based methods will get stuck at this infeasible solution.
>
> **Question 2: In Assumption 4.1, it is assumed that the Slater's condition holds. Does it mean that we cannot include the equality constraint $g(x)=0$ for the following analysis?**
>
> We can modify Assumption 4.1 to allow linear equality constraints like $\mathbf{A}\mathbf{x}=\mathbf{b}$ but not a nonlinear equality constraint like $g(\mathbf x)=0$. Under Assumption 4.1, the constraint is convex, but adding a nonlinear equality constraint will make the constraint non-convex, which will require a different analysis (e.g. the analysis in Section 5).
>
> Let me provide some details on how we can extend our algorithm and theory to allow linear equality constraints. More specifically, we can extend our results to the problem
>
> $\min_{\mathbf x\in\mathcal{X}}f(\mathbf x) \quad \text{s.t.} \quad h(\mathbf x)\leq 0,\quad \mathbf A\mathbf x=\mathbf b,$
>
> where $f$ is weakly convex and $h$ is convex. Then we still assume Assumption 4.1 except that 4.1B is changed to
>
> *B'. There exists $\mathbf x_{\text{feas}}\in\text{int}(\mathcal{X})$ such that $h(\mathbf x_{\text{feas}})<0$ and  $ \mathbf A\mathbf x_{\text{feas}}=\mathbf b$..*
>
> In this case, we only need to implement Algorithm 1 with $g(\mathbf x):=\max\\{h(\mathbf x), \\|\mathbf A \mathbf x-\mathbf b\\|_\infty\\}$. It is known that Slater's condition allows the present of linear equality constraints. This is why we only need $h$ to be strictly feasible.
> Using this fact, we can still obtain an upper bound of the Lagrangian multiplier similar to Lemma 4.2. Then Theorem 4.3 can be proved in the same way as before using this new upper bound. We will include this new result in the revision.

---

> > ### Comment · Reviewer_DHUQ · 2023-08-20
> >
> > Thank you for your clear and thorough response.

---

### Official Review · Reviewer_tsNX · 2023-07-07

**Soundness:** 3 good
**Presentation:** 3 good
**Contribution:** 3 good
**Rating:** 7
**Confidence:** 4

**Summary:**

The paper considers a weakly-convex constrained optimization problem and establishes the oracle complexity of the switching subgradient method for finding a nearly stationary point.

**Strengths:**

1. Considering a single-loop algorithm, the paper can establish the complexity $O(1/\epsilon^4)$ for a case of weakly-convex constraint. This is a nice result.

2. The paper has a rigorous convergence analysis.

**Weaknesses:**

In the numerical experiments, the paper compares a single-loop method with double-loop methods but reports the evolution of objective functions over iterations. This is not fair. The evolution over time should be reported.



**Questions:**

Can the analysis be extended to the case when Assumption 4.1 C is removed (for example, when $\mathcal X$ is unbounded)?



**Limitations:**

n.a.

---

> ### Author Rebuttal · Authors · 2023-08-09
>
> Thank you so much for reviewing our paper and providing valuable feedback that helps us further improve our work!
>
> **Comment: In the numerical experiments, the paper compares a single-loop method with double-loop methods but reports the evolution of objective functions over iterations. This is not fair. The evolution over time should be reported.**
>
> As the reviewer suggested, we have produced the figures showing how the objective value, infeasibility and stationarity measure evolve with CPU time in each algorithm. Please find the figures in the PDF file we submitted for this rebuttal. We will also include them in the revision of this manuscript.
>
> Comparing the plots in iterations and the plots in CPU time, we find that the curves of our SSG methods and the IPP-SSG method are almost unchanged, but the curve of the IPP-ConEx method becomes worse when plotted in CPU time. This is because the SSG and IPP-SSG methods only need to compute either $\mathbf{\zeta}_f^{(t)}$ or $\mathbf{\zeta}_g^{(t)}$ in each (inner) iteration while IPP-ConEx has to compute both $\mathbf{\zeta}_f^{(t)}$ and $\mathbf{\zeta}_g^{(t)}$ per inner iteration and thus induces additional computation cost. This suggests that the SSG methods we study are more efficient than the primal-dual methods like IPP-ConEx in computation time because the former computes fewer subgradients per iteration.
>
> **Comment: Can the analysis be extended to the case when Assumption 4.1 C is removed (for example, when $\mathcal{X}$ is unbounded)?**
>
> If we do not want to assume $\mathcal{X}$ is bounded, we can replace Assumption 4.1.C with the following assumption.
>
> *C'. There exists $D\in\mathbb{R}$ such that $\\|\mathbf{x}-\mathbf{x'}\\|\leq D$ for any $\mathbf{x}$ and $\mathbf{x'}$ in $\mathcal{S}:=\\{\mathbf{x}\in\mathcal{X}|g(\mathbf{x})\leq 0\\}$.*
>
> Here, $\mathcal{S}$ is the feasible set.  This assumption does not require a bounded $\mathcal{X}$. For example, when $\mathcal{X}=\mathbb{R}^d$, this new assumption means $g$ has a bounded $0$-sublevel set. One example satisfying this assumption is $g(\mathbf{x})=\\|\mathbf{A}\mathbf{x}-\mathbf{b}\\|_p-c\leq0$ where $\mathbf{A}$ has a full column rank and $p\geq 1$.  With this change to Assumption 4.1., we can still obtain the same complexity for the SSG method. We will include this new result in the revision.
>
> However, if we simply drop the boundedness assumption (Assumption 4.1.C) without making any alternative assumptions, we are unable to prove the same convergence results. This is because this boundedness assumption is critical for ensuring the Lagrangian multiplier $\widehat\lambda_t$ is bounded for all $t$ (see Lemma 4.2.) which is needed to prove the desired convergence rates. This boundedness assumption is also made in other papers on non-smooth non-convex constrained optimization, e.g., [14] and [58].

---

### Official Review · Reviewer_aPsS · 2023-07-13

**Soundness:** 3 good
**Presentation:** 3 good
**Contribution:** 3 good
**Rating:** 6
**Confidence:** 3

**Summary:**

This paper examines the oracle complexity of the switching subgradient method used for solving non-convex constrained optimization problems. The target functions are weakly convex, while the constraint functions are either convex or weakly convex. Remarkably, this method matches the complexity of double-loop methods while only necessitating a single loop implementation.



**Strengths:**

The authors have produced a commendable piece of work in the domain of weakly convex optimization, a specific category of nonconvex problems noted for its favorable theoretical convergence attributes. Although there is an abundance of single-loop algorithms such as subgradient or proximal point for weakly convex optimization, much of the existing literature has focused on unconstrained or simply-constrained issues. It is therefore somewhat unexpected that there are no analogous results in functionally constrained settings. Seen in this light, the technical novelty and contribution of this paper are remarkable.

**Weaknesses:**

While the authors have provided a comprehensive review of the literature, there seems to be an overemphasis on marginally relevant references. The broad coverage of optimization beyond weakly convex settings, although interesting, does not directly contribute to the key message of the paper.


**Questions:**

Assumption 4.1. assumes the boundedness of the solution, is this easily satisfied in practical scenarios?  Moreover,  should you include such a constraint in your second application?



Can you extend your algorithm for weakly convex problems with linear equality constraints?

---

> ### Author Rebuttal · Authors · 2023-08-09
>
> Thank you so much for reviewing our paper and providing valuable feedback that helps us further improve our work!
>
> **Comment: ... there seems to be an overemphasis on marginally relevant references. The broad coverage of optimization beyond weakly convex settings, although interesting, does not directly contribute to the key message of the paper.**
>
> In the revision, we will remove some marginally relevant references in the section of related works, for examples, the ones assuming smoothness, because weak convexity is more interesting when the problem is non-smooth.
>
> **Comment: Assumption 4.1. assumes the boundedness of the solution, is this easily satisfied in practical scenarios?**
>
> It depends on the applications. To train a real-world machine learning model, one can artificially add a ball constraint as a regularization technique to avoid overfitting. For example, in application (15) in Section 6.1, we add the constraint set $\mathcal{X}=\\{\mathbf
>  x\in\mathbb{R}^{d}|\\|\mathbf x\\|\leq r\\}$, which guarantees the boundedness of the feasible set. In practical scenarios, one can use cross validation to choose $r$ if necessary.
>
> If we do not want to assume $\mathcal{X}$ is bounded, we can replace Assumption 4.1.C with the following assumption.
>
> *C'. There exists $D\in\mathbb{R}$ such that $\\|\mathbf{x}-\mathbf{x'}\\|\leq D$ for any $\mathbf{x}$ and $\mathbf{x'}$ in $\mathcal{S}:=\\{\mathbf{x}\in\mathcal{X}|g(\mathbf{x})\leq 0\\}$.*
>
> Here, $\mathcal{S}$ is the feasible set. This assumption does not require a bounded $\mathcal{X}$. For example, when $\mathcal{X}=\mathbb{R}^d$, this new assumption means $g$ has a bounded $0$-sublevel set. One example satisfying this assumption is $g(\mathbf{x})=\\|\mathbf{A}\mathbf{x}-\mathbf{b}\\|_p-c\leq0$ where $\mathbf{A}$ has a full column rank and $p\geq 1$.  With this change to Assumption 4.1., we can still obtain the same complexity for the SSG method. We will include this new result in the revision.
>
> However, if we simply drop the boundedness assumption (Assumption 4.1.C) without making any alternative assumptions, we are unable to prove the same convergence results. This is because this boundedness assumption is critical for ensuring the Lagrangian multiplier $\widehat\lambda_t$ is bounded for all $t$ (see Lemma 4.2.) which is needed to prove the desired convergence rates. This boundedness assumption is also made in other papers on non-smooth non-convex constrained optimization, e.g., [14] and [58].
>
> **Comment: Moreover, should you include such a constraint in your second application?**
>
> We do not need to include such a constraint in the second application, i.e., problem (17). From the theoretical perspective, we have proved in Section B.1.1 in the appendix that (17) satisfies Assumption 5.1, which does not require a bounded $\mathcal{X}$, so we can just let $\mathcal{X}=\mathbb{R}^d$ and apply the convergence result in Theorem 5.6 based on Assumption 5.1. From the modeling perspective, (17) has already the regularization term $\text{SCAD}(\mathbf x)$ in the objective function. Hence, there is no need to add a ball constraint for the regularization purpose.
>
> **Comment: Can you extend your algorithm for weakly convex problems with linear equality constraints?**
>
> We consider two cases in this paper: (1) $g$ is convex; (2) $g$ is weakly convex. In the first case, we can extend our algorithm and theory to allow linear equality constraints. More specifically, we can extend our results to the problem
>
> $\min_{\mathbf x\in\mathcal{X}}f(\mathbf x) \quad \text{s.t.} \quad h(\mathbf x)\leq 0,\quad \mathbf A\mathbf x=\mathbf b,$
>
> where $f$ is weakly convex and $h$ is convex. Then we still assume Assumption 4.1 except that 4.1B is changed to
>
> *B'. There exists $\mathbf x_{\text{feas}}\in\text{int}(\mathcal{X})$ such that $h(\mathbf x_{\text{feas}})<0$ and  $ \mathbf A\mathbf x_{\text{feas}}=\mathbf b$..*
>
> In this case, we only need to implement Algorithm 1 with $g(\mathbf x):=\max\\{h(\mathbf x), \\|\mathbf A \mathbf x-\mathbf b\\|_\infty\\}$. It is known that Slater's condition allows the present of linear equality constraints. This is why we only need $h$ to be strictly feasible.
> Using this fact, we can still obtain an upper bound of the Lagrangian multiplier similar to Lemma 4.2. Then Theorem 4.3 can be proved in the same way as before using this new upper bound. We will include this new result in the revision.
>
>
> In the second case where $g$ is weakly convex, although we can still implement Algorithm 1 with the new $g$ defined above, we cannot extend the complexity theorem, i.e., Theorem 5.6, with linear equality constraints, unfortunately.

---

> > ### Comment · Reviewer_aPsS · 2023-08-18
> >
> > Thank you for the detailed reply. I will keep my rating.

---

### Author Rebuttal · Authors · 2023-08-09

We would like to thank all reviewers for reviewing our submission and providing valuable feedbacks. We have provided answers to each reviewer's questions separately. Reviewer tsNX suggested us to report the evolutions of the curves over time instead of over iterations. We have included the new figures in the uploaded PDF file with this rebuttal.

---

### Decision · Program_Chairs · 2023-09-21

**Decision:**

Accept (poster)

**Comment:**

All reviewers are positive about this paper and want to see it accepted. I went quickly over the paper and the main contributions myself and I find it interesting and thus further support acceptance. Please take into account the reviewers comments when preparing the final version.